# NETWORK OF THESEUS (LIKE THE SHIP)

**Vighnesh Subramaniam**[1*], **Colin Conwell**[2], **Boris Katz**[1],
**Andrei Barbu**[1], **Brian Cheung**[1*]
[1]MIT CSAIL, CBMM [2]Department of Cognitive Science, Johns Hopkins University
[1]{vsub851,boris,abarbu,cheungb}@mit.edu
[2]cconwel2@jhu.edu

## ABSTRACT

A standard assumption in deep learning is that the inductive bias introduced by a neural network architecture must persist from training through inference. The architecture you train with is the architecture you deploy. This assumption constrains the community from selecting architectures that may have desirable efficiency or design properties due to difficulties with optimization. We challenge this assumption with Network of Theseus (NoT), a method for progressively converting a trained, or even untrained, guide network architecture part-by-part into an entirely different target network architecture while preserving the performance of the guide network. At each stage, components in the guide network architecture are incrementally replaced with target architecture modules and aligned via representational similarity metrics. This procedure largely preserves the functionality of the guide network even under substantial architectural changes—for example, converting a convolutional network into a multilayer perceptron, or GPT-2 into a recurrent neural network. By decoupling optimization from deployment, NoT expands the space of viable inference-time architectures, opening opportunities for better accuracy–efficiency tradeoffs and enabling more directed exploration of the architectural design space.

## 1 INTRODUCTION

In machine learning research, we tend to assume that training is coupled with inference: If a specific architecture is trained to solve a task with specific inductive biases and computational mechanisms, that same architecture should be used at test time. This assumption is embedded in common practice: neural architecture search (NAS) discovers an efficient inference-time architecture that is then trained and deployed (Zoph & Le, 2016; Real et al., 2019; Liu et al., 2018). Compression pipelines prune and quantize a trained model to meet deployment budgets while preserving the trained structure (Han et al., 2015; Jacob et al., 2018). This assumption applies even in settings like distillation, where teacher networks train student networks designed explicitly for inference (Hinton, 2015; Gou et al., 2021; Romero et al., 2014; Huang et al., 2022).

We challenge this premise that the architecture used for training must be the architecture used for inference. Decoupling the architecture used for training from that used for inference would enable models to be trained with large, optimization friendly architectures and converted into lighter architectures for efficient inference on edge devices. This would also enable controlled exploration of inductive biases by comparing architectures without confounding optimization difficulty.

Large-scale analyses demonstrate that very different architectures (e.g., CNNs vs Transformers) converge toward similar internal representations as they scale (Huh et al., 2024; Han et al., 2023; Li et al., 2015; Conwell et al., 2024). Furthermore, classic observations of representational alignment across independently trained networks (Raghu et al., 2017; Kornblith et al., 2019) suggest that sufficiently expressive, distinct architectures can implement the same input–output functions. This statement of universality is also supported by theoretical work that shows that functions that are Turing-computable can be approximated by any neural network (Poggio & Fraser, 2024). However, the real challenge is not whether such functions exist, but how to reach them through optimization.

---

[*]Corresponding author.

This view implies that architectural priors are not constraints that must persist at inference, but are primarily training biases – scaffolds designed to guide and stabilize optimization.

Motivated by these perspectives and potential benefits, we introduce *Network of Theseus* (NoT), a part-by-part conversion procedure that starts from a given guide architecture and progressively replaces its components (e.g., layers) with target modules. Our name for this procedure is a reference to Plutarch's Ship of Theseus paradox (Plutarch, 100–125), which asks: once all the decaying planks of a ship are replaced (and none of the original remain), is it still the same ship? At each replacement stage, we instantiate a target module and optimize it to match the activations of the guide network using a representational distance function (Kornblith et al., 2019; Cristianini et al., 2001; Cortes et al., 2012). This optimization transfers priors between networks at a lower level, e.g., at the layer level rather than at the architectural level (Subramaniam et al., 2025). After the conversion is complete, the resultant target architecture is trained end-to-end on the downstream task. This alignment decouples training from deployment: we can train with one architecture and convert it into an entirely different one for test-time use, aiming to preserve the original performance as closely as possible.

Using NoT, we show broad architectural conversions: convolutional (CNN) models to fully connected MLPs with low-rank linear layers, vision transformers with multihead attention to token-wise MLPs, and transformer language models to Elman RNNs. Interestingly, we observe that even untrained guide networks contain useful inductive biases; NoT transfers these effectively, yielding performance comparable to using trained guides.

## 2   RELATED WORK

**Architectural Transfer via Linearizing Transformers**: NoT is a general method that builds on work applying cross-architectural distillation for linearizing transformers (Mercat et al., 2024; Zhang et al., 2024a;b; Bick et al., 2024). These works focus on specifically distilling multihead attention to either SSMs (Wang et al., 2024; Bick et al., 2025) or linear attention. Most approaches involve training linear attention layers or SSMs to approximate softmax attention via MSE. Some show that linearizing can specifically scale to very large transformers (Zhang et al., 2024a), e.g., 7B - 72B LLMs without considerable effect on performance.

**Model Distillation and Compression**: Distillation (Hinton, 2015; Gou et al., 2021; Sanh, 2019; Hsieh et al., 2023; Tian et al., 2019; Chen et al., 2021; Lin et al., 2020) transfers knowledge from a teacher model to a student model by introducing a new component to the loss function that forces the student model to behave like the teacher model (Kim et al., 2021; Zhou et al., 2021). This usually consists of penalizing the KL-divergence between the logit predictions of the student and teacher model. Methods have been proposed that use CKA as an alignment approach between representations of two networks or with representations in the brain, with notable improvement in network performance (Saha et al., 2022; Dapello et al., 2022). Other works also transfer knowledge across models while compressing information. For example, BERT-of-Theseus (Xu et al., 2020) uses the "Ship-of-Theseus" framing to compress BERT into a smaller architecture using stochastic replacement schedule during supervised training.

## 3   METHODS: NETWORK OF THESEUS (NOT)

**Network of Theseus (NoT)** provides a general procedure for replacing parts of a source ("guide") network with parts of a target architecture while preserving internal representations. At any time, a subset of components is replaced; we then train the replaced subset to *match* the guide's activations, stage by stage, according to a **replacement schedule**. NoT is task-agnostic. After the replacement completes, we finetune end-to-end for the downstream objective (e.g., image classification or language modeling).

Let $f^G(\cdot)$ be a fixed guide network with $k$ layers and let $f^T(\cdot; \theta)$ denote the under-construction target network obtained by replacing some of the guide's layers with new modules (e.g., Conv2d→Linear, attention variants, or block-level substitutions). We assume that both networks share input/output interfaces so that they can be run on the same inputs $x \in \mathbb{X}$. For a mini-batch from train, let $\boldsymbol{A}_i^G(\boldsymbol{x})$ and $\boldsymbol{A}_i^T(\boldsymbol{x}; \theta)$ be the activations extracted at layer $i$ from $f^G$ and $f^T$, respectively.

We write $g(\cdot, \cdot) \in [0, 1]$ for a representational similarity and use its complement $\Delta(\boldsymbol{A}, \boldsymbol{B}) \triangleq 1 - g(\boldsymbol{A}, \boldsymbol{B}) \in [0, 1]$ as a **dissimilarity** to be minimized. We instantiate $g$ with **linear CKA**.

For illustrative purposes, we show how to apply NoT for a single layer $i$. The matching loss is $\mathcal{L}_i(\boldsymbol{\theta}_i) = \mathbb{E}_{\boldsymbol{x} \sim \mathcal{D}} \left[ \Delta\left(\boldsymbol{A}_i^T(\boldsymbol{x}; \boldsymbol{\theta}_i), \boldsymbol{A}_i^G(\boldsymbol{x})\right)\right]$, where $\theta_i$ are the parameters introduced at location $i$ by the replacement. At a training *stage*, we optimize a set $I \subseteq \{1, \cdots, k\}$ of replaced layers jointly: $\mathcal{L}_I(\boldsymbol{\theta}_I) = \mathbb{E}_{\boldsymbol{x} \sim \mathcal{D}} \left[ \frac{1}{|I|} \sum_{i \in I} \Delta\left(\boldsymbol{A}_i^T(\boldsymbol{x}; \boldsymbol{\theta}_I), \boldsymbol{A}_i^G(\boldsymbol{x})\right)\right]$. Layers not in $I$ are frozen to prevent drift in the guide pathway.

**Replacement Modules**: NoT is agnostic to the specific replacement, requiring only shape compatibility. For Conv2d, we use an *equivalent low-rank linear* that applies to a flattened input and reshapes back to the expected output. Because of a large parameter count for the target linear layer, we first apply a linear layer to project the input to a lower dimensionality and then use another linear layer to project the desired output dimensionality, both of which are tunable. For attention, we support replacements with basic tokenwise MLPs, without any token-mixing, or RNNs. The framework also accommodates block-level replacements e.g., ResNet-50→ ResNet-18 basic blocks.

## 4 EXPERIMENTS

We apply NoT across a range of architectures and replacements to measure how well we can use our staged replacement across different replacement schedules. For all experiments, we compare using a trained and untrained guide network and separate training for representational similarity and task training into two stages.

**Tasks**: We consider image classification and language modeling in this paper. For testing image classification, we use the ImageNet dataset (Deng et al., 2009), measuring Top-1 performance on the pre-defined validation set. For language modeling, we use the Wikitext-103 dataset (Merity et al., 2016) where models must predict the next token given some context. We use a sequence length of 128 for all models and use the training, validation and testing sets defined by the dataset. We tokenize the text using the GPT-2 (Radford et al., 2019) tokenizer.

**Architectures and Replacements**: We study cross-family and within-family conversions spanning convolutions, attention, and recurrent computation. In all cases, layers are progressively replaced and optimized to match the guide's intermediate activations via representational similarity. *ResNet-18→MLP*: We convert ResNet-18 (He et al., 2016) to a fully-connected MLP by replacing each convolutional layer with a linear layer. This removes the spatial priors of convolutional layers, such as receptive fields or translational equivariance. We use two low-rank linear layers (rank 256-1024) to keep the parameter size of the linear layers reasonable, as discussed in Section 3. *DINOv2→Patch-MLP*: We convert DINOv2 (Oquab et al., 2023) to a patch-wise MLP by replacing each multihead attention with a token-wise MLP that applies feedforward layers independently to each visual token. This eliminates cross-token communication, increasing efficiency at the cost of token interaction. *GPT-2→RNN*: We replace GPT-2 (Radford et al., 2019; Vaswani, 2017) multihead attention layers with two Elman RNN layers per attention block. This tests whether sequential memory-based processing can substitute for parallel attention mechanisms.

**Training**: For each setting, we train NoT with a guide network and the target replacements. We include a baseline with *naive replacement* where all guide network layers are replaced with the target modules from scratch for a fair comparison to understand what representational similarity provides to performance. The naive replacement is run for the same total number of gradient steps as NoT to ensure fair comparison. During task training, all networks are trained with cross-entropy loss, without loss of generality. For all training, we use a consistent batch size of 256. Furthermore, we employ different learning rates across the progressive stages of representational similarity. We apply task training for 100 epochs across all settings. All networks across all settings are trained with 5 random seeds to compute error bars. Our error bars are associated with the standard error across all seeds.

## 5 RESULTS

We apply NoT to all of our previously described networks and summarize the results in Table 1. We find that across all settings, NoT improves performance over naive replacement by up to 30%. For example, we find that fully-connected MLPs can only be trained to achieve 33% accuracy. With

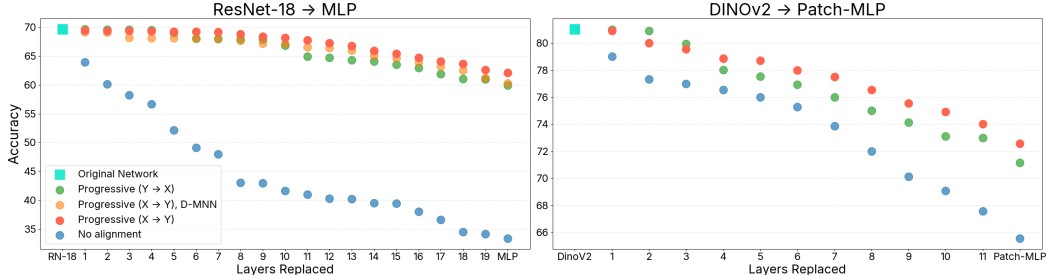

Figure 1: **Progressive layer replacement preserves performance across replacements.** We visualize progressive layer replacement across all patches. We apply a patch, reduce the CKA and finetune the resultant hybrid network until full replacement. This is compared with naive replacement with no CKA alignment. We compare forward replacement (X→Y, reverse replacement (Y→X) in both settings, and compare using D-MNN in the ResNet-18→MLP setting. Across all progressive replacements, we far exceed naive replacement with no alignment. We are not sensitive to replacement order.

| Original (Guide) → Target | Similarity Metric | Guide | NoT (Trained) | NoT (Untrained) | Baseline |
|---|---|---|---|---|---|
| | | **ImageNet Top-1 Accuracy** ($\uparrow$) | | | |
| ResNet-18 → MLP | CKA | 69.66 | 62.12±0.42 | 60.85±0.48 | 40.01±1.44 |
| DINOv2 → Patch-MLP | CKA | 81.03 | 72.56±0.20 | 70.39±0.25 | 68.72±0.55 |
| | | **Wikitext-103 Perplexity** ($\downarrow$) | | | |
| GPT-2 → RNN | CKA | 37.50 | 50.58±0.89 | 58.26±0.43 | 119.44±3.01 |

Table 1: **Network of Theseus vastly improves over naive replacement and preserves performance**. We compare NoT with standard training (baseline) and the original guide network performance. We find that NoT vastly outperforms naive replacement, close to 30% on ImageNet and 71 points on Wikitext-103. NoT also shows vast improvement even when the the guide is untrained.

NoT, we improve by 30% and identify a fully-connected MLP that is competitive with ResNet-18. Similarly, we are able to replace attention in vision transformers with Patch-MLPs while preserving accuracy, meaning that token communication is not necessary for downstream image classification, achieving competitive results on DINOv2. This holds for Elman RNNs, which become effective replacements for attention computations. Most excitingly, we find that NoT can be applied across similar architectures like ResNet-18. These results far exceed naive replacement.

Additionally, in Figure 1, we show training performance across our progressive layer replacements for ResNet-18 and DINOv2. We show the original performance, training with NoT and standard training with a naive replacement. When replacing with NoT, we consider progressive replacement in both the forward direction, from the first layer to the last layer, and the reverse direction, from the last layer to the first layer. This tests how sensitive results are to replacement direction. Across all layer replacements, we find that using representational similarity to incorporate a layer leads to stronger results in comparison to standard training. Reverse replacement leads to slightly worse performance, likely due to representational similarity drift. The improvement is significant across all layer replacements. We find that performance is lost on certain layers such as layer 13 in ResNet-18, showing that these layers are bottlenecks.

Surprisingly, we find that untrained guide networks are able to transfer useful architectural priors via NoT, leading to similar improvements. This is shown in Table 1. Across all settings, we see improvements with NoT even when the guide network is completely untrained. This shows that untrained networks have useful priors as noted in previous work (Subramaniam et al., 2025; Ulyanov et al., 2018; Zhong & Andreas, 2024). This distinguishes NoT from distillation given that distillation does not work with untrained networks. More importantly, we believe this has striking implications for architecture transfer. We highlight our result with ResNet-18 and ResNet-50. Converting from an untrained ResNet-50 to ResNet-18 results in a 3% performance increase. This suggests that depth and connectivity can act as transferable priors even without learned weights. We believe this has strong implications beyond NoT for distillation, where the assumption was that we would always need a trained architecture to train another trained architecture.

# 6 CONCLUSION

Training and inference are usually coupled, but Network of Theseus (NoT) breaks this coupling by progressively replacing a guide network with target modules while aligning intermediate representations; therefore, the trained function can be carried into a different inference architecture. Across ImageNet and Wikitext-103, this staged conversion preserves much of the guide's performance for large cross-family changes, specifically ResNet-18→MLP, DINOv2→Patch-MLP, GPT-2→RNN, whereas naive replacement collapses. Therefore, representational alignment is the key mechanism.

We believe NoT reframes how we think about architectural design. If representational alignment can be used to carry performance across network architecture families, then the new goal of an architecture during training is to construct discoverable representations rather than dictating the deployed inference architecture.

**Limitations**: We have not fully explored the space of architectures in our current work. This lack of coverage is due to computational limitations as the progressive process requires more resources that simply fitting all layers at once.

## ACKNOWLEDGMENTS

This work was supported by the DARPA Mathematics for the DIscovery of ALgorithms and Architectures (DIAL) program, the DARPA Knowledge Management at Scale and Speed (KMASS) program, the NSF award 2124052, the MIT-IBM Watson AI Lab, the Department of the Air Force Artificial Intelligence Accelerator under Cooperative Agreement Number FA8750-19-2-1000, and the Air Force Office of Scientific Research (AFOSR) under award number FA9550-21-1-0399. The views and conclusions contained in this document are those of the authors and should not be interpreted as representing the official policies, either expressed or implied, of the Department of the Air Force or the U.S. Government. The U.S. Government is authorized to reproduce and distribute reprints for Government purposes notwithstanding any copyright notation herein.

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
