# OpenReview forum: "Network of Theseus (Like the ship)"
_ICLR.cc/2026/Workshop/Sci4DL — Sci4DL 2026_

### Official Review · Reviewer_3RXP · 2026-02-24

**Fit:** 2
**Significance:** 2
**Confidence:** 2

**Summary:**

This paper proposes Network of Theseus (NoT), a progressive architecture conversion framework that challenges the conventional coupling between training-time and inference-time architectures. Instead of assuming that the model architecture optimized during training must also be used at deployment, the authors introduce a staged replacement procedure that gradually transforms a “guide” network into a structurally different “target” architecture. During each stage, newly introduced target modules are optimized to match the intermediate representations of the guide network via a representational similarity metric (linear CKA). Once the replacement is complete, the fully converted model is fine-tuned on the downstream task.

The method is evaluated across several cross-family conversions, including ResNet-18 to MLP, DINOv2 to patch-wise MLP, and GPT-2 to Elman RNN, on ImageNet and Wikitext-103. Empirical results demonstrate that NoT significantly outperforms naive direct replacement and preserves a substantial fraction of the guide network’s performance, even under drastic architectural changes. An additional notable observation is that untrained guide networks can still transfer useful inductive biases through representational alignment.

**Strengths:**

The primary contribution of this work lies in its conceptual reframing of architectural design. By decoupling optimization architecture from deployment architecture, the paper introduces a novel perspective that may expand the design space for efficient inference models. This shift in viewpoint is original and potentially impactful, particularly in contexts where optimization-friendly architectures differ substantially from deployment-efficient ones.

Methodologically, the framework is simple and broadly applicable. The reliance on representational alignment rather than logit-level distillation enables transfer across heterogeneous model families. The use of CKA as a similarity metric is well grounded in prior literature on representational analysis. The experimental results consistently demonstrate large improvements over naive replacement across both vision and language tasks, including challenging cross-family transformations such as Transformer-to-RNN conversion.

The finding that untrained guide networks can provide useful structural priors is particularly intriguing. This result suggests that architectural inductive bias alone may be sufficient to scaffold effective representations, which has broader implications for understanding the role of structure in deep learning.

**Suggestions:**

Despite its conceptual appeal, the work would benefit from deeper theoretical justification. While representational alignment empirically enables cross-architecture transfer, the paper does not clearly articulate the conditions under which such alignment is sufficient to preserve function-level behavior. A more principled discussion, potentially in terms of representational equivalence or functional approximation, would strengthen the contribution.

The computational overhead of the progressive replacement procedure is also not thoroughly analyzed. Since NoT involves multiple stages of alignment and optimization, it may incur significantly higher training costs than directly training the target architecture. Without reporting total compute or training time comparisons, it is difficult to assess practical feasibility.

The experimental baselines could be stronger. The primary comparison is against naive replacement, which is expected to perform poorly in drastic cross-family conversions. Including comparisons to stronger alternatives such as representation-level distillation, logit distillation, joint training, or low-rank adaptation methods would provide a more comprehensive evaluation.

---

### Official Review · Reviewer_K7Vm · 2026-02-24

**Fit:** 2
**Significance:** 2
**Confidence:** 2

**Summary:**

The paper proposes Network of Theseus (NoT), a method that decouples the neural architecture used during training and inference. NoT replace blocks from the guide network architecture (training) with different fine-tuned blocks to build the target network, preserving performance.

**Strengths:**

- Structure: The paper is well-written and easy to follow
- Motivation: The paper addresses important topics such as changing the network during inference, but I have some concerns (see below)

**Suggestions:**

- Motivation and experiments: The motivation is clear, for example, using a more lightweight network during inference. However, authors mainly show how to transform a residual CNNs or transformers to MLPs or RNNs. Authors should discuss the increased computation during inference (number of parameters, etc.)
- Method: How NoT fits in the literature of layer-wise distillation?

---

### Official Review · Reviewer_TYwy · 2026-02-27

**Fit:** 2
**Significance:** 1
**Confidence:** 2

**Summary:**

The authors posit that the Deep model architecture is an optimisation scaffold and the actual model used in practice need not follow from the one trained. this is significant conceptual contribution. They outline an algorithm to replace a guide architecture with a target architectures. Support this empirically with experiments that show 1. this has merit even with untrained models 3. several guide-target pairs show significant performance increase.

While the paper presents an intriguing direction, at present it needs additional evidence (some claims are at odds with empirical results) and contextualisation.

**Strengths:**

The paper test their theory across a number of domains (e.g. vision, language, etc).
Clear experimental setup and procedures.
The experiment with untrained models is the strongest supporting evidence, but needs to be better qualified e.g. impact of initialisation (if any), generalisation, etc

**Suggestions:**

- The paper claims that there is not degradation of performance however the table seems to be at odds e.g.69.22 -> 62.85, 36.50 -> 58.26.
- more comprehensive experiments are needed.
- greater discussion of relationship with distillation and novelty over previous work (e.g. it appears that this extends previous work to dissimilar architectures)
- greater discussion of the costs/complexity associated (any speed up, cost/impact of CKA, etc)

---

### Meta-Review · Area_Chair_rnTY · 2026-03-01

**Recommendation:** Accept

**Metareview:**

Recommend borderline acceptance.

A technical note: it's not clear to me, on an initial read, if (a) the amount of compute is properly controlled between experimental settings and (b) sufficiently strong baselines are used.

---

### Decision · Program_Chairs · 2026-03-02

Accept